# The Therapeutic Potential of ADAMTS8 in Lung Adenocarcinoma without Targetable Therapy

**DOI:** 10.3390/jpm12060902

**Published:** 2022-05-30

**Authors:** Hsiao-Chen Lee, Chao-Yuan Chang, Kuan-Li Wu, Hung-Hsing Chiang, Yung-Yun Chang, Lian-Xiu Liu, Yung-Chi Huang, Jen-Yu Hung, Ya-Ling Hsu, Yu-Yuan Wu, Ying-Ming Tsai

**Affiliations:** 1Graduate Institute of Medicine, College of Medicine, Kaohsiung Medical University, Kaohsiung 807, Taiwan; hc890131@gmail.com (H.-C.L.); chaoyuah@kmu.edu.tw (C.-Y.C.); 980448kmuh@gmail.com (K.-L.W.); ji394122@gmail.com (L.-X.L.); beryl1992@gmail.com (Y.-C.H.); yainghsu@kmu.edu.tw (Y.-L.H.); fred901229@gmail.com (Y.-Y.W.); 2Division of Plastic Surgery, Department of Surgery, Kaohsiung Medical University Hospital, Kaohsiung Medical University, Kaohsiung 807, Taiwan; 3Department of Anatomy, College of Medicine, Kaohsiung Medical University, Kaohsiung 807, Taiwan; 4Division of Pulmonary and Critical Care Medicine, Kaohsiung Medical University Hospital, Kaohsiung 807, Taiwan; cyy807@gmail.com (Y.-Y.C.); jyhung@kmu.edu.tw (J.-Y.H.); 5Division of Thoracic Surgery, Department of Surgery, Kaohsiung Medical University Hospital, Kaohsiung Medical University, Kaohsiung 807, Taiwan; shiiiiidae@gmail.com; 6Division of General Medicine, Kaohsiung Medical University Hospital, Kaohsiung 807, Taiwan; 7Department of Internal Medicine, Kaohsiung Municipal Ta-Tung Hospital, Kaohsiung 807, Taiwan; 8Department of Medical Research, Kaohsiung Medical University Hospital, Kaohsiung 807, Taiwan; 9Drug Development and Value Creation Research Center, Kaohsiung Medical University, Kaohsiung 807, Taiwan; 10School of Medicine, College of Medicine, Kaohsiung Medical University, Kaohsiung 807, Taiwan

**Keywords:** ADAMTS8, EGFR, GATA1, NKT, lung cancer, Treg, PD-L1

## Abstract

Lung cancer is well known for its high mortality worldwide. The treatment for advanced lung cancer needs more attention to improve its survival time. A disintegrin and metallopeptidase with thrombospondin motifs 8 (ADAMTS8) has been linked to several cancer types. However, its role in lung cancer is worthy of deep investigation to promote novel drug development. This study took advantage of RNA-seq and bioinformatics to verify the role that ADAMTS8 plays in lung cancer. The functional assays suggested that ADAMTS8 mediates invasion and metastasis when expressed at a low level, contributing to poor overall survival (OS). The expression of ADAMTS8 was under the regulation of GATA Binding Protein 1 (GATA1) and executed its pathologic role through Thrombospondin Type 1 Domain Containing 1 (THSD1) and ADAMTS Like 2 (ADAMTSL2). To define the impact of ADAMTS8 in the lung cancer treatment strategy, this study further grouped lung cancer patients in the TCGA database into mutated epidermal growth factor receptor (EGFR)/wild-type EGFR and programmed death ligand 1 (PD-L1) high/low groups. Importantly, the expression of ADAMTS8 was correlated positively with the recruitment of anticancer NKT cells and negatively with the infiltration of immunosuppressive Treg and exhausted T cells. The results indicated that lung cancer patients with higher ADAMTS8 levels among wild-type EGFR or low PD-L1 groups survive longer than those with lower levels do. This study indicates that ADAMTS8 might be a treatment option for patients with lung adenocarcinoma who lack efficient targeted or immunotherapies.

## 1. Introduction

Lung cancer is a leading cause of cancer-related death worldwide due to smoking, asbestos and radon exposure [1,2]. To cure lung cancer would be a challenging issue for today’s medicine. The treatment strategy for lung cancer includes traditional chemotherapy, radiotherapy and operation [3,4]. The discovery of druggable targets such as epidermal growth factor receptor (EGFR), anaplastic lymphoma receptor tyrosine kinase (ALK), MET, ROS-1receptor tyrosine kinase and Kirsten rat sarcoma viral oncogene homolog (KRAS) open avenues for treating lung cancer [4,5]. In addition, the new strategy of boosting tumor-killing immunity via PD-1/PD-L1 and CTLA4 has advanced the treatment of lung cancer [6]. Despite modern standard therapies such as targeted therapy (EGFR tyrosine kinase inhibitors) and immune checkpoint inhibitors, the likelihood of a cure is low, with low 5-year overall survival [7]. Moreover, the emergence of resistance makes the situation worse [8,9]. To address this critical issue, more efforts are needed to discover other possible druggable targets to overcome lung cancer.

A disintegrin and metallopeptidase with thrombospondin motifs (ADAMTSs) are a group of 19 members functioning as secreted, multi-domain and extracellular zinc metalloproteinases in humans [10]. These 19 members are grouped as proteoglycanases or aggrecanases (ADAMTS1, 4, 5, 8, 9, 15 and 20), the procollagen N-propeptidases (ADAMTS2, 3 and 14), the cartilage oligomeric matrix protein-cleaving enzymes (ADAMTS7 and 12), the von-Willebrand Factor proteinase (ADAMTS13) and a group of orphan enzymes (ADAMTS6, 10, 16, 17, 18 and 19) based on their biologic functions [10]. In addition, there are seven ADAMTS-like proteins (ADAMTSL1-6 and papilin) in different species, such as humans, mice and other mammals [11]. The structure of ADAMTS proteases comprises an N-terminal protease domain with catalytic activity and a C-terminal ancillary domain. This indicates that ADAMTS proteases have a similar catalytic mechanism to Matrix MetalloProteinases (MMPs) and A Disintegrin and Metalloproteinases (ADAMs), using a Zn-atom coordinated by three conserved His residues [12,13]. However, the ADAMs enable protein ectodomain shedding from the cell surface [13], but ADAMTS proteases are secreted out into the extracellular matrix (ECM) to mediate proteolytic events in the ECM. When being secreted, ADAMTS proteins appear to have a great propensity to bind to the cell surface and/or pericellular matrix and undergo extensive post-translational modification, including disulfide bonding and N- and O-glycosylation [14].

ADAMTS genes play roles in a wide range of physiological processes, including cellular functions such as cell proliferation, apoptosis, migration, invasion, angiogenesis and extracellular matrix degradation [15]. The functional modification might be affected by post-transcriptional alternative splicing and post-translational mechanisms, as well as furin-mediated protease activation [8]. The clinical manifestations of Mendelian disorders resulting from mutations in ADAMTS2, ADAMTS10/19, ADAMTS13, ADAMTS17 and ADAMTS18 are Ehlers–Danlos syndrome [16], Weil–Marchesani syndrome 1 [17], thrombotic thrombocytopenic purpura [18], Weil–Marchesani-like syndrome [19] and microcornea, myopic chorioretinal atrophy and telecanthus (MMCAT), respectively. The Mendelian diseases could be modified by concurrent or compensating expression and functional overlap of a homolog [20]. Other than these Mendelian disorders, they are also involved in a variety of diseases, including osteoarthritis and malignant tumors. Growing evidence indicates that the abnormal expression of ADAMTSs plays an important role in human tumorigenesis, progression and metastasis. They play a dual role in cancer; for instance, ADAMTS14 works as a promoter of hypermethylation and acts as a tumor suppressor gene, whereas others show upregulation in cancer acting as oncogenes, including ADAMTS12, which regulated the cell proliferation and migration in colorectal cancer [21,22]. However, their role in lung cancer remains unclear. Next-generation sequencing (NGS), a powerful tool, has been developed in recent years to elucidate the mechanisms of tumorigenesis and devise a possible solution with the aid of bioinformatics [23]. Taking advantage of clinical specimens and NGS with the aid of bioinformatics, this study has pointed out that ADAMTS8 might be involved in lung cancer pathogenesis. It would be an option to treat lung cancer not harboring driver mutations or not legible for immune checkpoint inhibitor (ICI) therapy.

## 2. Materials and Methods

### 2.1. NGS and Quantitative Real-Time Reverse Transcription Polymerase Chain Reaction (Q-RT-PCR) Assay

Eight pairs of adjacent lung non-tumor and tumor tissues were collected from the Division of Thoracic Surgery and Division of Pulmonary and Critical Care Medicine, Kaohsiung Medical University Hospital (Kaohsiung, Taiwan). The protocol was reviewed and approved by the Institutional Review Board of Kaohsiung Medical University Hospital (KMUH-IRB-20130054; KMUH-IRB-20180023). The deep sequencing for the tissues was performed by a biotechnology company, Welgene (Taipei, Taiwan). Samples were prepared using the Illumina sample preparation kit according to the TruSeq Small RNA Sample Preparation Guide. The 3′ and 5′ adaptors were ligated to total RNA, and reverse transcription was followed by PCR amplification. The enriched cDNA constructs were size-fractionated and purified via 6% polyacrylamide gel electrophoresis. Libraries were sequenced on an Illumina instrument (75SE cycle single read) and processed using the Illumina software. Data were analyzed by Feature Extraction 10.7.3.1 software, and raw signal data were normalized by quantile normalization for differentially expressed gene discovery. The criteria for differentially expressed mRNA by NGS analysis were fold change > 2 and fragments per kilobase million (FPKM) > 0.3.

Total RNA from the cells was isolated using TRIzol (Life Technologies, Waltham, MA, USA). cDNA was prepared using an oligo (dT) primer and reverse transcriptase (Takara, Shiga, Japan), following standard protocols. RNA levels were determined using real-time analysis with SYBR Green on a StepOne-Plus machine (Applied Biosystems, Foster City, CA, USA). The relative expression of mRNA in cells was normalized to GAPDH. The relative standard method (2^−ΔΔCt^) was used to calculate relative RNA expression. The following primers were used: ADAMTS8 (forward, 5′- AGCCAAGTACCAGTCATGCC-3′ and reverse, 5′- CTCGGCAGAACAACTTGCAG-3′) and GAPDH (glyceraldehyde 3-phosphate dehydrogenase) (forward, 5′-TTCACCACCATGGAGAAGGC-3′ and reverse, 5′-GGCATGGACTGTGGTCATGA-3′).

### 2.2. Cell Culture

Human lung cancer PC-9 cells and HBE135-E6E7 (CRL-2741™) were obtained from the European Collection of Cell Cultures (ECACC) and American Type Culture Collection (ATCC), respectively. Human lung adenocarcinoma (LUAD) cell lines CL1-0 and CL1-5 were kindly provided by Pan-Chyr Yang (National Taiwan University, College of Medicine). The CL1-0, CL1-5 and PC-9 were cultured in RPMI 1640 medium (Cat. No. 12-702Q, Lonza (Walkersville, MD, USA)) supplemented with 10% fetal bovine serum (FBS) (EMD Millipore, Billerica, MA, USA), 2 mM glutamine and 1% penicillin–streptomycin (Cat. No. 17-745E, Lonza, Walkersville, MD, USA). HBE135 cells were cultured in a keratinocyte serum-free medium with 5 ng/mL EGF, 0.05 mg/mL bovine pituitary extract (Invitrogen, formerly GIBCO-BRL, Cat. No. 17005-042), 0.005 mg/mL insulin (Cat. No. I3536, Sigma-Aldrich, St Louis, MO, USA) and 500 ng/mL hydrocortisone (Cat. No. H0135, Sigma-Aldrich). All cell lines were tested for mycoplasma contamination using mycoplasma test kits (Mycoalert Mycoplasma Detection Kit, Cat. No. LT07-318) (Lonza) every 3 months.

Total RNA from the cells was isolated using TRIzol (Life Technologies). cDNA was prepared using an oligo (dT) primer and reverse transcriptase (Cat. No. RR037A, Takara, Shiga, Japan), following standard protocols. RNA levels were determined using real-time analysis with SYBR Green (Cat. No. A25742, Thermo Fisher Scientific, USA) on a StepOne-Plus machine (Applied Biosystems, Foster City, CA, United States). The relative expression of mRNA in cells was normalized to GAPDH. The relative standard method (2^−ΔΔCt^) was used to calculate relative RNA expression. The primers were determined from the well-known Primer 3 tool. The following primers were used: ADAMTS8 (forward, 5′- AGCCAAGTACCAGTCATGCC-3′ and reverse, 5′- CTCGGCAGAACAACTTGCAG-3′) and GAPDH (glyceraldehyde 3-phosphate dehydrogenase) (forward, 5′-TTCACCACCATGGAGAAGGC-3′ and reverse, 5′-GGCATGGACTGTGGTCATGA-3′).

### 2.3. Bioinformatics

Gene expression quantification datasets of lung adenocarcinoma (LUAD) samples were extracted from the Oncomine database (http://www.oncomine.org, Compendia biosciences, Ann Arbor, MI, USA) [24] and TCGA (https://portal.gdc.cancer.gov/, (accessed on 1 September 2021)). The criteria in the analysis were fold change > 2 and *p*-value < 0.05, which was calculated using Oncomine or UALCAN (http://ualcan.path.uab.edu/, (accessed on 1 September 2021)) [25]. The correlation of specific genes and survival rates in lung cancer was assessed by the KM plotter (http://kmplot.com/analysis/, (accessed on 1 September 2021)) [26]. Patients were divided into 2 groups with the best cut-off, which was computed with median survival. The hazard ratios (95% confidence intervals) were calculated using the Cox proportional model. The analysis of protein–protein interactions was conducted using the STRING database (https://string-db.org/, (accessed on 1 September 2021)) [27]. CancerSEA was used to elucidate the functions of ADAMTS8 (http://biocc.hrbmu.edu.cn/CancerSEA/home.jsp, (accessed on 29 March 2022)) [28]. The distribution and infiltration of various immune cells were obtained from ImmuCellAI (Immune Cell Abundance Identifier) (http://bioinfo.life.hust.edu.cn/ImmuCellAI/#!/document, (accessed on 29 March 2022)) [29].

### 2.4. Transcription Factors of ADAMTS8

Potential transcription factors of ADAMTS8 were curated from the website hTFtarget (http://bioinfo.life.hust.edu.cn/hTFtarget#!/, (accessed on 11 January 2022)) [30]. The survival significance of each transcription factor was determined via the KM plotter website, whereas the expression level of TF was extracted from the TCGA LUAD cohort. The transcription factors downregulated in the tumor part with a hazard ratio (HR) for overall survival (OS) less than 1 were presumed as the activators of ADAMTS8. In contrast, the transcription factors upregulated in tumors with a HR for OS of more than 1 were considered to be the repressors for ADAMTS8. The correlation of the expression level between ADAMTS8 and each potential transcription factor curated from the above process was tested in the TCGA LUAD cohort by Pearson’s method.

### 2.5. Screening for Differentially Expressed miRNAs

TargetScan (https://www.targetscan.org, (accessed on 11 January 2022)) and miRDB (http://mirdb.org/, (accessed on 11 January 2022)) were used to predict the candidate miRNAs that could regulate the expression of ADAMTS8. The miRNAs predicted by the two databases simultaneously were selected as the objects for further analysis.

### 2.6. Gene Set Enrichment Analysis

Gene set enrichment analysis (GSEA) is a computational tool that evaluates whether an a priori defined gene set has statistically significant, concordant differences between two biological or pathological states. To investigate the role of ADAMTS8 in LUAC, the patients of the TCGA were divided into ATAMTS8 high expression and low expression groups, and GSEA was conducted to analyze the enrichment of datasets between the high-ADAMTS8 and low-ADAMTS8 groups. False discovery rate (FDR) < 0.005 and nominal *p*-value < 0.001% were set as the cut-off criteria.

### 2.7. The Survival Significance of ADAMTS8 in EGFR-Mutant or EGFR-Wild-Type Tumors

The somatic mutation profile of EGFR was extracted from the TCGA LUAD dataset. Those samples with common EGFR mutations, including exon 19 deletion or exon 21 L858R, were defined as the EGFR-mutant group (*n* = 35). Those without mutation detected on EGFR were defined as the EGFR-wild-type group (*n* = 445). Uncommon and compound mutations of EGFR were excluded and discarded from the analysis (*n* = 29). The survival significance of ADAMTS8 was determined by comparing the survival time between the ADAMTS8 high and low subgroups in the EGFR-mutant and -wild-type groups in the TCGA database, respectively. The survival difference between subgroups was tested by a log-rank test.

### 2.8. Validation of ADAMTS8 on Databases of Lung Cancer Cell Lines

The expression of ADAMTS8 in primary and metastatic adenocarcinoma cell lines with or without EGFR mutation was obtained from the Cancer Cell Line Encyclopedia (CCLE) database (https://sites.broadinstitute.org/ccle/, (accessed on 29 March 2022)) [31].

### 2.9. Statistical Analysis

Data are expressed as mean ± standard deviation (SD). All data were analyzed using GraphPad Prism 9.2.0 (GraphPad Software, La Jolla, CA, USA), and differences between the means for each condition were evaluated by a one-way ANOVA followed by a Tukey post hoc test. Pearson’s correlation between ADAMTS8 and miRNAs or mRNAs in the TCGA LUAD dataset was calculated.

## 3. Results

The expression of ADAMTS8 is low in lung adenocarcinoma. To investigate the role of ADAMTS family genes in lung cancer, we first assessed the levels of various ADAMTS genes from eight paired normal and tumor tissues undergoing the bulky NGS in our in-house cohort as flow charts (Figure A1). The result showed that the expression of different functional groups of ADAMTS, including the polymerization of extracellular matrix (ECM) proteins, catalyzation of proteoglycan degradation and others (family), did not differ except for ADAMTS7 and ADAMTS8 (Appendix A and Figure 1A). However, the *p*-value of ADAMTS7 was 0.049 and overall survival was not significant between low and high expression. To further assess the impact of ADAMTS8, we utilized the data of the TCGA cohort and discovered a lower level of ADAMTS8 in tumor parts (Figure 1B) and other members of the ADAMTS family, except for ADAMTS10 (Appendix A). With the utilization of the public web resource UALCAN, the expression levels of ADAMTS8 were found to be different between normal and tumor parts but not lymph node metastasis or stage-dependent (Figure 1C,D). In addition, four different cohorts, including Landi, Okayama, Selamat, Su and Hou, also showed that ADAMTS8 was downregulated in tumor parts in lung cancer patients (Figure 1E). We also found that CL1-5 expressed a lower level of ADMATS8 than CL1-0 (less invasive) (Figure 1F). The results suggested that ADAMTS8 expression was reduced in lung cancer.

Patients with lower expression levels of ADAMTS8 survive for a shorter duration in lung adenocarcinoma. To investigate whether ADAMTS8 functions as an antitumor gene in lung cancer, survival analysis of ADAMTS8 would assign it a significant clinical role. The study utilized the public website of the KM plotter to assess this. First, the results suggested that the lower expression of ADAMTS8 led to a worse survival rate for 3 out of 4 different cohorts in terms of the overall survival (OS) (Figure 2A). Moreover, the lower expression of ADAMTS8 caused a shorter time for first progression (FP) in 2 out of 3 different cohorts (Figure 2B). On the contrary, concerning post-progression survival (PPS), the levels of ADAMTS8 did not show any significant impact (Figure 2C). These data suggested that ADAMTS8 could be a potential prognostic marker for lung cancer.

Low-expressed ADAMTS8 potentiates more aggressive behaviors in lung adenocarcinoma. To evaluate the role of ADAMTS8 in lung cancer proregression, we performed GSEA to predict the cancer behavior by comparing the transcriptomes of ADAMTS8 in lung cancer patients with higher and lower expression in the TCGA database. GSEA results showed that decreased ADAMTS8 was associated with poor survival, epithelial–mesenchymal transition (EMT) and metastasis (Figure 3A–C). The data retrieved from the dedicated public database, CancerSEA, revealed several functional roles, such as angiogenesis, apoptosis, cell cycle, differentiation, DNA damage, DNA repair, EMT, inflammation, invasion and metastasis in glioblastoma multiforme (GBM), retinoblastoma and uveal melanoma (Figure 3D). When considering GBM, the relationships between ADAMTS8 and invasion, EMT or metastasis were negatively correlated, with r values of −0.4, −0.35 or −0.31, respectively (Figure 3E–G). These bioinformatics data suggest that low-expressed ADAMTS8 is associated with a more aggressive phenotype in lung adenocarcinoma.

The DNA methylation, copy number variation and miRNA did not affect the expression of ADAMTS8. To elucidate the possible mechanisms for ADAMTS8 dysregulation in lung cancer, we assessed several epigenetic regulatory mechanisms, such as DNA methylation, copy number variation (CNV) and miR-RNA interaction. First, when utilizing the public cohort of the TCGA from UALCAN, the methylation promotion of ADAMTS8 did not differ in the tumor, lymph node metastasis or tumor stages compared with normal parts (Figure 4A). In addition, there was no significant correlation between the expression of *ADAMTS8* mRNA and *ADAMTS**8* CNV in lung adenocarcinoma in the TCGA cohort (Figure 4B). Furthermore, to elucidate the miR-ADAMTS8 interaction, the miRNAs were predicted by databases miRDB_6.0 (target score > 90) and TargetScan_8.0 (conserved sites, context++ score percentile > 90), and there were 11 candidate miRNAs (Appendix A), but only miR 98-5p had a significantly negative correlation (r = −0.33) with the expression of ADAMTS8 in our cohort and the TCGA database (Figure 4C,D). However, the level of miR-98-5p was not different between tumor and normal parts in the TCGA database or in lymph node metastasis and staging (Figure 4E). Moreover, the expression of miR-98-5p did not affect the OS of lung cancer patients, suggesting that miR-98-5p is not the main regulatory mechanism of ADAMTS8 (Figure 4F). These results do not indicate that the expression of ADAMTS8 is under the aforementioned epigenetic regulations.

GATA1 regulated ADAMTS8 expression. Transcription factors (TFs) act as the main regulators to turn the specific target genes on or off by attaching binding elements to promoter regions. This study utilized hTFtarget to identify several TFs that could bind the specific regions of the ADAMTS8 promoter, including GATA1, RARA, LMO2, E2F4 and CTBP2 (Table 1). Among them, GATA1, RARA and LMO2 were lowly expressed, while E2F4 and CTBP2 were highly expressed in tumor parts. However, only GATA1 and LMO2 showed more than a two-fold change between normal parts and tumor parts in lung cancer (Table 1). The expression levels of GATA1 and LMO2 were low and associated with advanced lymph node metastasis and staging in lung cancer (left two panels, Figure 5A,B). Lower GATA1 and LMO2 were associated with a shorter overall survival time (right panel, Figure 5A,B). Cross-analysis of GATA1-ADAMTS8 in overall survival time showed that low-level ADAMTS8 caused a shorter survival rate than high-level ADAMTS8 regardless of GATA1 levels (Figure 5C). In contrast, there was no survival benefit of ADAMTS8 based on low-level but not high-level LMO2 in lung cancer patients (Figure 5D). These data suggested that the interaction of ADAMTS8 and GATA1, but not LMO2 by transcriptional regulation, could affect the clinical outcome of lung cancer.

The molecular network of THSD1, ADAMTSL2 and ADAMTS8. Signaling transduction pathways manipulate cell growth/apoptosis, and aberrant pathways might cause diseases. To delineate the signaling transduction pathway for ADAMTS8, STRING (protein, version 11.0b) was used as a tool to predict the interaction network of ADAMTS8. There were ten proteins interacting with ADAMTS8 (Figure 6A), including ADAMTS2, ADAMTS10, ADAMTS13, ADAMTS19, ADAMTSL2, THBS1, THBS2, THSD1, ACAN and POFUT2. Among them, eight proteins were excluded based on low correlations with ADAMTS8. Only two proteins, THSD1 and ADAMTSL2, correlated well with ADAMTS8 (Table 2). The correlations of THSD1 and ADAMTSL2 with ADAMTS8 amounted to 0.75 and 0.58, respectively (Figure 6B,C). Low expression of THSD1 existed in the tumor parts, lymph node metastasis and tumor stage compared with the normal parts in lung cancer patients (Figure 6D). Low expression of ADAMTSL2 was presented in the tumor parts but was not related to lymph node metastasis and tumor stage compared with the normal parts (Figure 6E). A poor overall survival time was also found in patients with lower expression of ADAMTSL2 and THSD1 (Figure 6F,G). The results would explain the protein interactions mediating lung cancer tumorigenesis.

High-level ADAMTS8 conferred survival benefits in patients lacking the PD-L1 and targeted gene. The treatment strategies for lung cancer have expanded in the past decade. However, lung cancer patients with no driver mutations or negative PD-L1 expression lack definite treatment options other than chemotherapy [4]. The discovery of a new treatable target would benefit lung cancer patients without eligible targeted agents or immunotherapies. The results showed that the levels of ADAMTS8 were lower in the lower PD-L1 (CD274) expression patient group (left panel, Figure 7A). The survival advantage of ADAMTS8 existed only in patients with low PD-L1 (right panel, Figure 7A) and not in patients with high PD-L1 (middle panel, Figure 7A). At the same time, the expression of ADAMTS8 was not different between high and low levels of CTLA4 in lung cancer patients, and the benefits of higher ADAMTS8 expression were present regardless of the levels of CTLA4 (Figure 7B). Moreover, regarding the targetable gene, EGFR, the survival advantage of ADAMTS8 existed only in wild-type EGFR and not in common mutations (right lower panel, Figure 7C). More importantly, the expression of ADAMTS8 was positively correlated with anticancer NKT and negatively with immune-suppressive regulatory T cells (Table 3). The immediate conclusion is that patients with low-level PD-L1 and wild-type EGFR might benefit from high levels of ADAMTS8. To further verify the impact of ADAMTS8 on the target therapies of EGFR-mutant lung cancer, we assessed the levels of ADAMTS8 in several LUAD cell lines with mutated EGFR based on the sensitivity of TKI treatment. As shown in Figure 7D, cancer cell lines including HCC827 (*del E746*-*A750* in exon 19), HCC2279 (*del E746*-*A750* in exon 19) and HCC4006 (*del E746*-*A750* in exon 19) have higher levels of *ADAMTS8* expression than in TKI-resistant cancer cell lines (H1650 (*del E746*-*A750* in exon 19) and H1975 (*L858R* in exon 21)).

## 4. Discussion

Nowadays, curing lung cancer is a major challenge for medical oncologists and pulmonologists and an important prospect for patients worldwide. The advanced treatments with EGFR TKIs and ICIs have enabled lung cancer treatment to achieve a step forward. However, these solutions are not suitable for lung cancer patients not bearing driver mutations or high PD-L1—for example, as a treatment strategy for triple-negative breast cancer. This study utilized the precious clinical specimens sequenced by NGS and Big Data analysis via bioinformatics to elucidate a possible candidate gene, ADAMTS8, and provide a potential strategy for patients who have not benefited from TKIs or ICIs in lung cancer treatment.

Members of the ADAMTS family share common structural features, and their aberrant expression has led to different human diseases. ADAMTS8 belongs to the ADAMTS family, also known as METH-2, which play roles in a wide range of physiological processes, such as cell proliferation, apoptosis, migration, invasion, angiogenesis and extracellular matrix degradation [32]. It was initially recognized as an antiangiogenic factor in a variety of diseases, such as pulmonary artery hypertension [32] and different neoplastic diseases [33,34]. This study utilized bulky NGS to analyze the expression of different members of the ADAMTS family from eight paired normal and tumor tissues. The expression levels of ADAMTS8 were lower in tumor parts in our clinical cohorts, the TCGA and seven other cohorts, but the relationships were not lymph node metastasis or stage-dependent. Moreover, several lung cancer cell lines have been investigated, and the results showed lower levels of ADAMTS8 compared with normal bronchial epithelial cells (HBE 135). To further validate the role of ADAMTS8 clinically, the survival analyses from KM plotter revealed a longer overall survival time (OS) and time to first progression (FP) but not post-progression survival (PPS) in lung cancer patients with high ADAMTS8 expression. ADAMTS8, but not other members of the ADAMTS family, plays a role in mediating lung cancer.

Carcinogenesis requires several oncogenic activities, such as the inhibition of DNA repair and apoptosis and the enhancement of migration, invasion, EMT, angiogenesis, etc. [35]. When utilizing the database from the CancerSEA, we found that low levels of ADAMTS8 enabled cancer cell invasion in glioblastoma multiforme (GBM). This invasion phenotypic change was consistent with breast cancer [33]. The GSEA using the TCGA cohort also suggested that ADAMTS8 was involved in poor survival in the Shedden lung cancer cohort. Moreover, ADAMTS8 favored EMT in GBM. Different cell lines, such as esophageal and nasopharyngeal cell lines, also favored their role in EMT [36]. The GSEA suggested the role of ADAMTS8 in EMT in breast cancer. Finally, concerning metastasis, ADAMTS8 mediates metastasis in GBM. This result is consistent with hepatocellular carcinoma [37]. The GSEA data favored the role of ADAMTS8 in enhancing the metastasis of melanoma. ADAMTS8 promotes angiogenesis in RB and PAH. However, a report from Zhang et al. stated that ADAMTS8 suppressed lung cancer progression by suppressing VEGFA [38].

Transcription factors regulate the specific gene expression to exert its designated function. The derived data from “hTFtarget” showed that curated transcription factors (TFs) regulate the expression of ADAMTS8. There were five TFs linked to the expression of ADAMTS8. Only two out of three lowly expressed TFs, namely GATA1 and LMO2, exhibited more than a two-fold change in tumor parts. GATA1 contains two highly conserved zinc fingers, one of them being located at its C-terminal. ADAMTS8 is under the regulation of GATA1. GATA1 has been linked to survival time in lung cancer patients [39]. This study further validated that the low expression of GATA1 means shorter survival in lung cancer. When cross-analyzing ADAMTS8 based on the levels of GATA1 in survival time, lung cancer patients with low-level ADAMTS8 survived for a shorter period when compared with high-level based on low GATA1 but not LMO2 expression. In a previous study, DNA methylation lowered the expression of ADAMTS8 to promote gastric cancer carcinogenesis [40]. However, this mechanism did not explain the low levels of ADAMTS8 in the cohort of the TCGA in lung cancer, or the lymph node metastasis and tumor stage. Further investigation of the copy number variation of ADAMTS8 did not support this epigenetic regulatory mechanism in colorectal cancer linked to ADAMTS4 [41]. The related protein network from the Pathway Commons suggested that ADAMTSL2 and THSD1 were correlated with the expression of ADAMTS8. These two proteins conferred a survival disadvantage when expressed at low levels.

Curing lung cancer remains a major challenge in modern medicine. However, novel strategies such as the targeted therapy of EGFR TKIs and immune checkpoint inhibitors, as alternatives to chemotherapy, bring hope for lung cancer patients. However, these treatments cannot meet all the requirements. More efforts are necessary to satisfy the unmet need. In lung cancer patients with low expression levels of PD-L1, who might not benefit from ICI therapy, they survived for shorter times when they possessed lower levels of ADAMTS8 compared to patients with higher levels. In addition, in lung cancer patients with wild-type EGFR, lower levels of ADAMTS8 might be of a survival benefit compared with higher levels. This might imply that lung cancer patients ineligible for treatment with ICIs or EGFR TKIs might benefit from ADAMTS8 treatment. Our findings revealed ADAMTS8 as a potential prognostic marker in lung cancer. Low ADAMTS8 expression may contribute to a shorter survival duration and be a promising target for lung cancer patients who are ineligible for TKI or ICI treatment.

## 5. Conclusions

In conclusion, ADAMTS8 plays a role in lung cancer carcinogenesis and immunity. ADAMTS8 might play a role in lung adenocarcinoma without targetable medication.

## Figures and Tables

**Figure 1 jpm-12-00902-f001:**
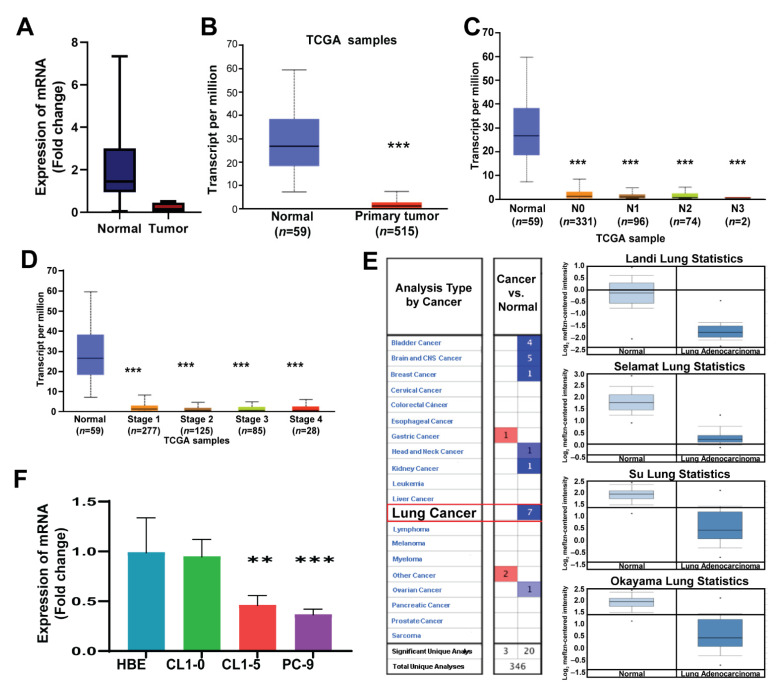
The expression of ADAMTS8 is low in lung adenocarcinoma. The roles of ADAMTS8 in lung cancer were investigated. The lower expression levels of ADAMTS8 from tumor parts in our in-house cohort (**A**) and patients from TCGA cohort (**B**). With the utilization of the public data of TCGA, the expression of ADAMTS8 was observed to be lower in lymph node metastasis (**C**) and advanced stage (**D**) but not lymph node metastasis or stage-dependent. In addition, the datasets from Oncomine showed lower expression levels of ADAMTS8 in tumor parts among seven cohorts (**E**). The expression of ADAMTS8 in normal and lung cancer cell lines (**F**). Significant difference between cancer and normal parts. ** *p* < 0.01, *** *p* < 0.005.

**Figure 2 jpm-12-00902-f002:**
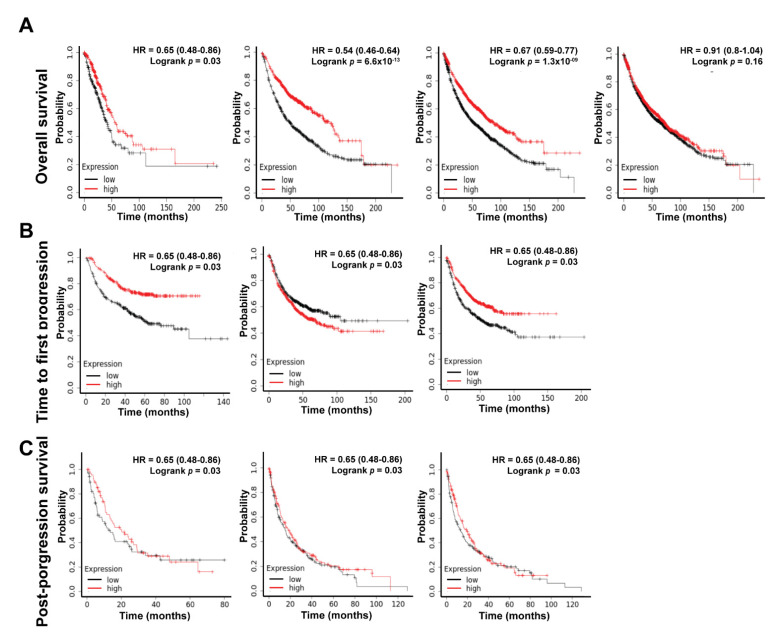
Patients with lower expression levels of ADAMTS8 survive for a shorter duration in lung adenocarcinoma. ADAMTS8 had low expression in tumor parts, and survival analyses were performed to confirm whether ADAMTS8 plays a significant clinical role. The KM plotter revealed that the lower expression of ADAMTS8 conferred lower overall survival (**A**) and shorter time for first progression (**B**) but not post-progression survival (**C**). Abb.: OS, overall survival; FP, time to the first progression; PPS, post-progression survival.

**Figure 3 jpm-12-00902-f003:**
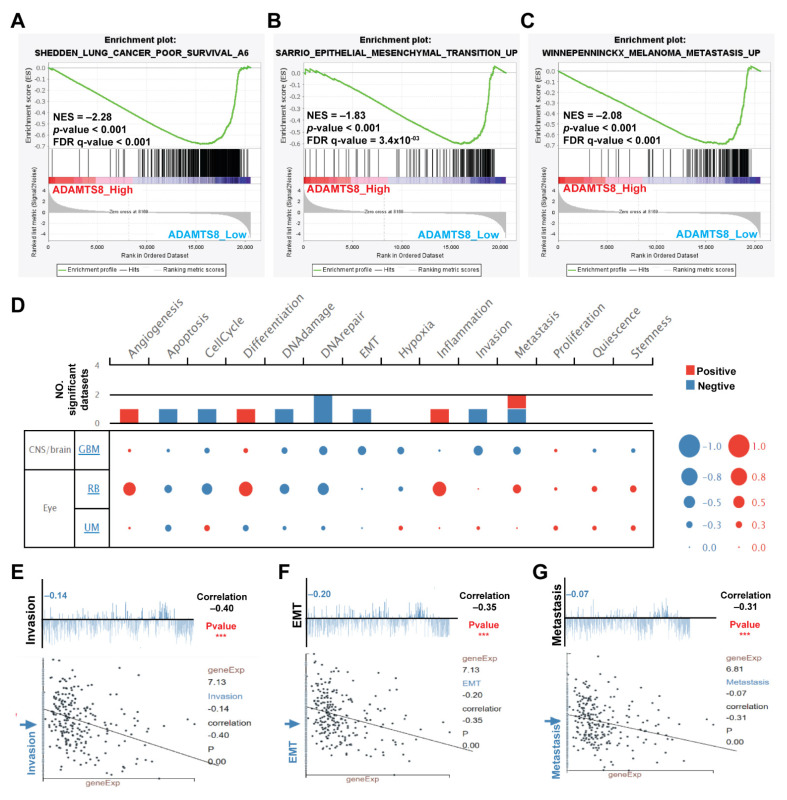
ADAMTS8 caused a more aggressive phenotype in lung adenocarcinoma. The GSEAs linked to low levels of ADAMTS8 were lung cancer, poor survival, EMT and metastasis in cancer (**A**–**C**). From the public database CancerSEA, ADAMTS8 mediates several cancer cellular behaviors, with positive correlations in angiogenesis, differentiation and inflammation and negative correlations in apoptosis, cell cycle, DNA damage, DNA repair, EMT, invasion and metastasis (**D**). The correlations in invasion, EMT and metastasis yielded values of −0.4, −0.35 and −0.31, respectively (**E**–**G**). Abb. EMT, epithelial-mesenchymal transition; GBM, glioblastoma multiforme; RB, retinoblastoma; UM, uveal melanoma. *** *p* < 0.005.

**Figure 4 jpm-12-00902-f004:**
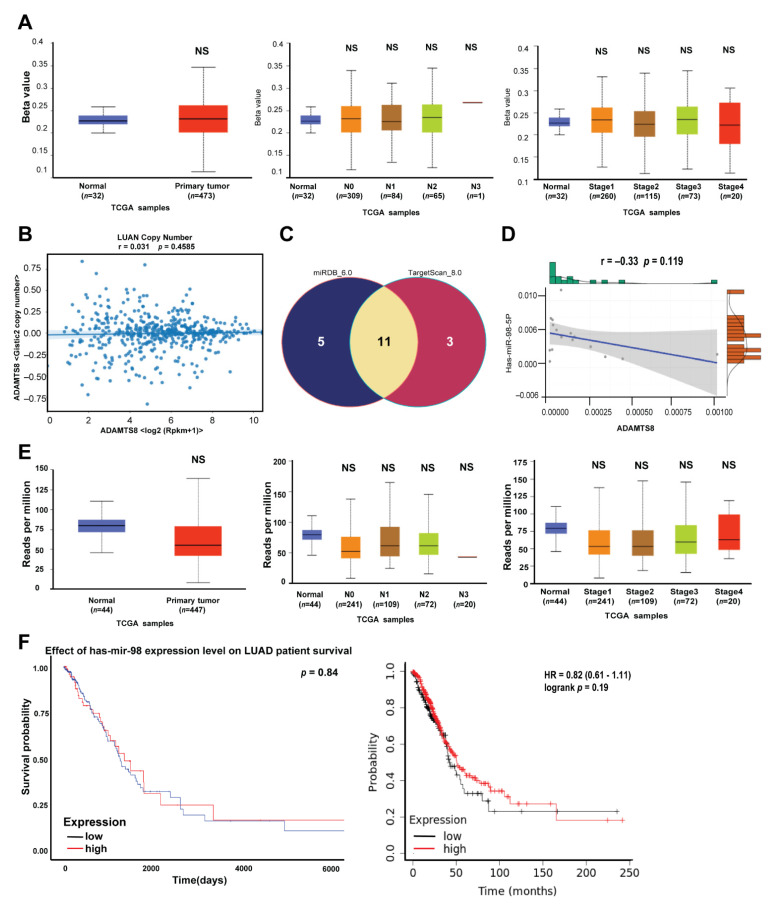
The possible regulatory factors affecting the expression of ADAMTS8. The levels of DNA methylation expressed in normal/tumor tissues (left panel, **A**), lymph node metastasis (middle panel, **A**) and tumor stage (right panel, **A**). The correlation between *ADAMTS8* mRNA expression and *ADAMTS8* copy number variation (**B**). The predicted miRNAs from miRDB_6.0 and TargetScan_8.0 (**C**) and the correlation between miR-98-5p and *ADAMTS8* (**D**). The expression levels of miR-98-5p between normal parts and tumor parts, in lymph node metastasis and tumor staging (**E**), and survival analysis (**F**) in lung cancer. NS, not significant.

**Figure 5 jpm-12-00902-f005:**
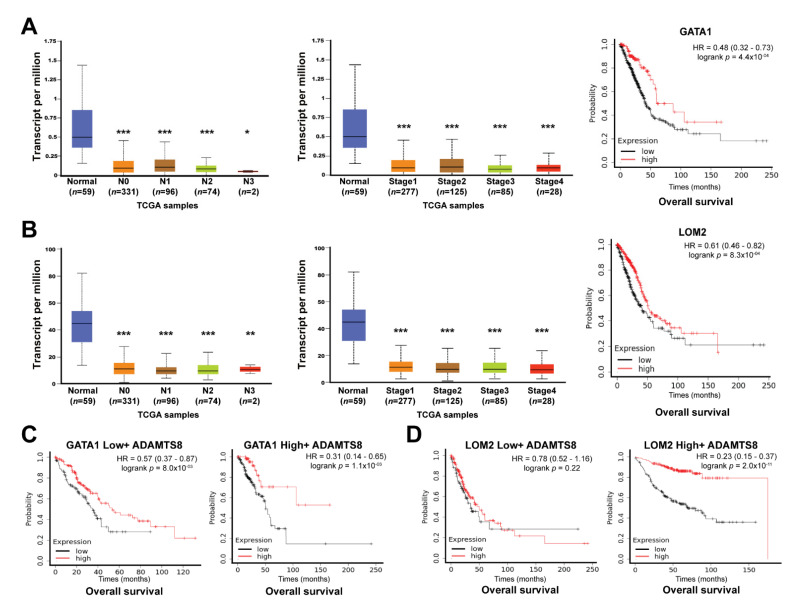
GATA1 regulated *ADAMTS8* expression and conferred poor survival. The GATA1 expression in lymph node metastasis and tumor staging (left two panels, **A**) and its survival disadvantage (right panel, **A**). The LMO2 expression in lymph node metastasis and tumor staging (left two panels, **B**) and its survival disadvantage (right panel, **B**). The cross-analysis of effects of ADAMTS8 on overall survival analysis in lung cancer patients based on either GATA1 (**C**) or LMO2 (**D**). * *p* < 0.05, ** *p* < 0.01, *** *p* < 0.005.

**Figure 6 jpm-12-00902-f006:**
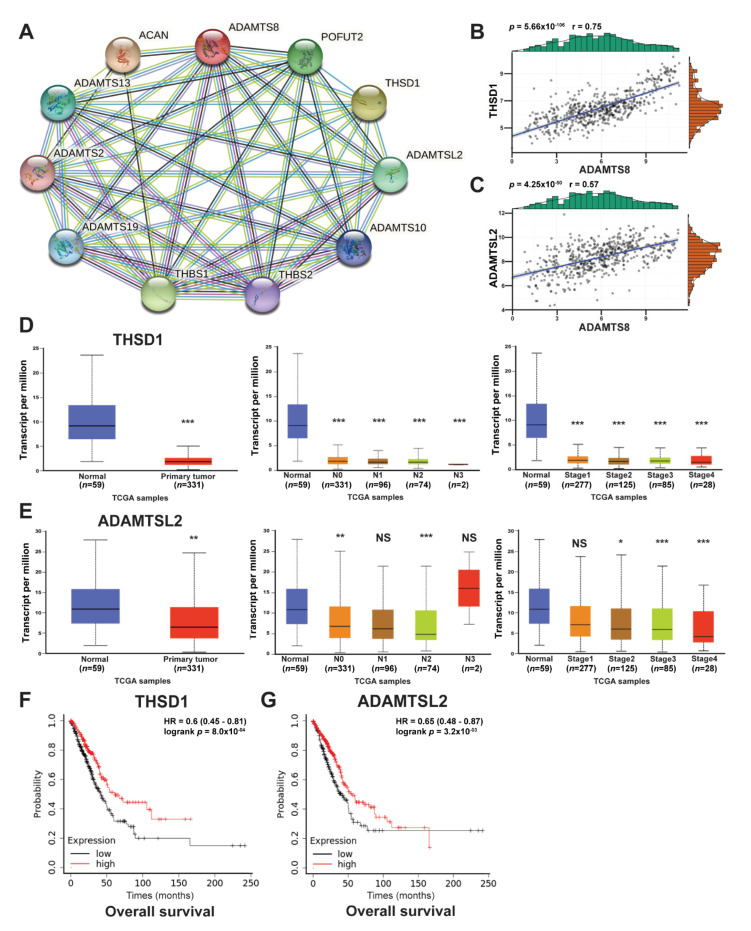
The network interaction between ADAMTSL2, THSD1 and ADAMTS8. The protein–protein interactions of ADAMTS8 utilizing STRING (protein, version 11.0b) revealed the interactions (**A**). The r values for the correlation between ADAMTS8 and ADAMTSL2 and THDS1 were 0.58 and 0.75, respectively, for the TCGA cohort (**B**,**C**). The expression of ADAMTSL2 in the tumor parts was lower than in the normal parts (left panel, **D**); we also show its relation with lymph node metastasis or tumor stage (middle and right panels, **D**). The expression of THSD1 in tumor parts was lower than in normal parts (left panel, **E**); we also show its relation with lymph node metastasis or tumor stage (middle and right panel, **E**). The effects of THSD1 (**F**) and ADAMTSL2 (**G**) on overall survival analysis. * *p* < 0.05, ** *p* < 0.01, *** *p* < 0.005.

**Figure 7 jpm-12-00902-f007:**
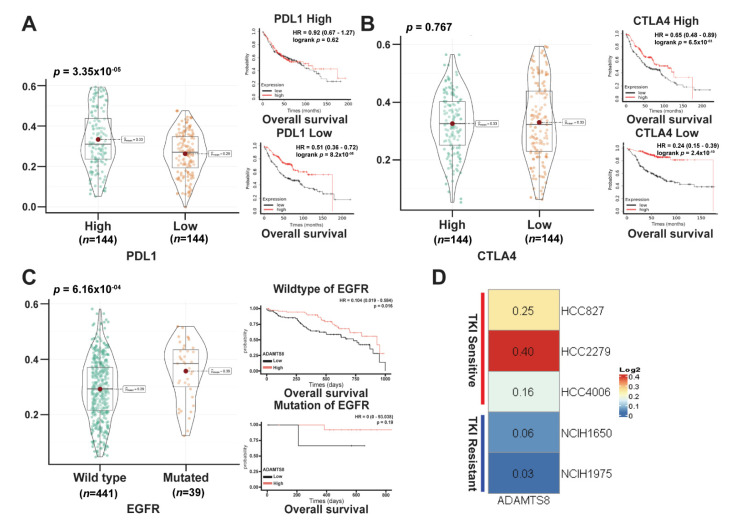
Patients with low PD-L1 expression or no driver mutations would benefit from ADAMTS8 treatment. The expression levels of ADAMTS8 are listed based on PD-L1 (CD274) expression (left panel, **A**). The survival analysis of high versus low ADAMTS8 expression divided by high and low expression of PD-L1 showed a benefit of ADAMTS8 in patients with low PD-L1 but not high PD-L1 (mid, right panels, **A**). CTLA4 showed a survival benefit regardless of levels of expression in CTLA4 (**B**). Regarding a targetable gene, EGFR, patients harboring the wild-type mutations but not common mutations with high-level ADAMTS8 survived longer (**C**). The expression of ADAMTS8 in various LUAD cancer cell lines (**D**).

**Table 1 jpm-12-00902-t001:** Survival significance of curated co-association transcription factors and their correlations with targeted *ADAMTS8.*

	TF Expression Level ^a^	TF Expression Level ^b^	Correlation with ADAMTS8 ^b^	
Transcription factors	Log2 Fold Change	Log2 Fold Change	Pearson’s r	*p*-value
Low expression in tumor				
**GATA1**	−0.85	−1.09 *	0.62	3.96 × 10^−63^
**RARA**	0.33	−0.01 *	0.19	4.13 × 10^−6^
**LMO2**	−1.25	−0.26 *	0.59	3.27 × 10^−54^
High expression in tumor				
**E2F4**	0.24	0.12 *	−0.28	7.89 × 10^−12^
**CTBP2**	0.32	0.14 *	−0.35	2.75 × 10^−18^

^a^. Data of the transcription factor expression level and its correlation with ADAMTS8 were extracted from our in-house cohort. ^b^. Data of the transcription factor expression level and its correlation with ADAMTS8 were extracted from the TCGA LUAD cohort. * *p* < 0.001.

**Table 2 jpm-12-00902-t002:** The potential interactive molecules of ADAMTS8.

	Survival Association ^a^		mRNA Expression Level ^b^	Correlation with ADAMTS8 ^b^	
mRNA	HR for OS *	*p*-Value	Log2 Fold Change	Pearson’s r	*p*-Value
**POFUT2**	0.58	5.6 × 10^−10^	1.11	−0.12	3.00 × 10^−3^
**ACAN**	0.57	3.7 × 10^−11^	1.44	−0.17	4.50 × 10^−5^
**ADAMTS19**	0.60	3.1 × 10^−9^	0.73	0.03	4.06 × 10^−1^
**THBS1**	0.68	7.1 × 10^−6^	0.99	0.19	3.39 × 10^−6^
**ADAMTS13**	0.60	1.4 × 10^−9^	1.12	0.11	1.00 × 10^−2^
**ADAMTS2**	0.66	1.0 × 10^−6^	1.04	0.06	1.54 × 10^−1^
**ADAMTS10**	0.60	1.5 × 10^−9^	1.04	0.25	9.61 × 10^−10^
**THBS2**	0.81	1.4 × 10^−2^	1.36	−0.36	1.29 × 10^−18^

^a^. Hazard ratio of overall survival was extracted from the KM plotter website. ^b^. Data of the mRNA expression level and its correlation with ADAMTS8 were extracted from the TCGA LUAD cohort. * OS, overall survival.

**Table 3 jpm-12-00902-t003:** TCGA LUAD immune cell correlations with ADAMTS8.

Immune Cells	Pearson’s r	*p*-Value
NKT	0.630	2.50 × 10^−64^
Tfh	0.510	2.39 × 10^−39^
NK	0.450	1.02 × 10^29^
Th2	0.440	1.09 × 10^−28^
CD4_T	0.400	1.36 × 10^−23^
MAIT	0.400	3.76 × 10^−23^
Tgd	0.280	8.47 × 10^−12^
Macrophage	0.190	2.53 × 10^−6^
Tc	0.130	2.00 × 10^−3^
Th17	0.009	8.25 × 10^−1^
CD4_naive	−0.001	9.79 × 10^−1^
DC	−0.020	6.20 × 10^−1^
Neutrophil	−0.020	6.79 × 10^−1^
Tcm	−0.030	5.39 × 10^−1^
CD8_T	−0.050	2.59 × 10^−1^
CD8_naive	−0.060	1.27 × 10^−1^
B_cell	−0.080	5.20 × 10^−2^
Tr1	−0.110	9.00 × 10^−3^
iTreg	−0.200	8.80 × 10^−7^
Monocyte	−0.260	1.70 × 10^−10^
Th1	−0.290	1.92 × 10^−12^
Tex	−0.390	5.26 × 10^−22^
Tem	−0.420	3.48 × 10^−26^
nTreg	−0.670	3.11 × 10^−77^

*Abb.* NKT, natural killer T cell; Tfh, follicular helper T cell; MAIT, mucosal-associated invariant T cell; Tgd, gamma delta T cell; Tc, cytotoxic T cell; DC, dendritic cell; Tcm, central memory T cell; Tr1, type 1 regulatory T cell; iTreg, induced Treg; Tex, exhausted T cell; Tem, effector memory T cell; nTreg, natural Treg.

## Data Availability

CancerSEA (http://biocc.hrbmu.edu.cn/CancerSEA/home.jsp), CCLE database (https://sites.broadinstitute.org/ccle/), hTFtarget (http://bioinfo.life.hust.edu.cn/hTFtarget#!/), ImmuCellAI (http://bioinfo.life.hust.edu.cn/ImmuCellAI/#!/document), KM plotter (http://kmplot.com/analysis/), miRDB (http://mirdb.org/), STRING database (https://string-db.org/), TargetScan (https://www.targetscan.org), TCGA (https://portal.gdc.cancer.gov/), UALCAN (http://ualcan.path.uab.edu/, all accessed on 16 May 2022.

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
