# Peer review of "The Therapeutic Potential of ADAMTS8 in Lung Adenocarcinoma without Targetable Therapy"

_jpm, 2022, doi:10.3390/jpm12060902_

Round 1
Reviewer 1 Report
Lee et al willing to investigate the role of ADMATS8 in the pathogenesis of lung cancer. The study has several experimental, intellectual and presentational weakness that contribute to the overall poor quality of the scientific product.
Before commenting the content of the paper, we need to make some comments on the language, structure and figure quality of the manuscript because both interfere with the ability to comprehend what the authors are willing to say. To begin with, the use of english language through out the manuscript is inadequate at best. There is a great need of language editing since it contains numerous grammatical errors and typos (way too many examples to add here) and in certain places is pretty unclear what the authors are willing to express. Furthermore, the preparation and quality of the figures is low, blurry, overstretched and not clearly labeled through out the manuscript that someone has to be constantly back and forth the figure legend in order to try to comprehend the message. On a different note, the legends are not informative and repetitive. In addition, the structure of the paper with large gaps between the figures is very confusing and tiring to read. It seems like a collection of paragraphs and pages that non-coherently connect figures, making it difficult to follow.
If someone can overcome these issues, that largely introduce increasing challenges on the understanding of the manuscript's content, there are several flows with the methodology itself. The authors constantly ignore, cherry-pick and move back and forth between different subtypes of lung cancer (LUAD, LUSC etc) without proposing or demonstrating why (figures 1,2,3 Given that these molecular/clinical subtypes are crucial for the prognosis and management of lung cancer patients, this is a severe flaw of the story, which is not justified within the text or the data.
In figure 1F, the authors present data on the expression of ADAMTS8 in several lung cancer cells (ps in the methods for some reason they present PC-9, a lung adenocarcinoma cell line with EGFR mutation, as a lung cancer cell line, but the other cell lines as LUAD ones. The authors are not using widely used LUAD (A549, H1299, H1975 etc) or LUSC cell lines to support their claims. Furthermore, nowadays, there are several depositories with RNA-seq expression of several cancer cell lines (especially from the broad institute) that the authors could use, in order to provide more unbiased and robust data. In figure 3D, I find extremely challenging to understand why the authors include data regarding GBM and other CNS tissues. Also in the version that I downloaded, figure 4 is missing.
Moving on, data on Figure 5 demonstrate that GATA1 and LMO2 affect the prognosis of LUAD patients (again not defined why LUAD or other subtype), trying to make a connection with DAMTS8 regulation. These data are purely non-informative and correlative. The fact that these widely regulatory TFs affect the prognosis has nothing to do with ADAMTS8 expression. To address this claim, the authors should perform a plethora of biological experiments with genetic knockdowns, mouse models etc.
In a similar trend, I have increasing challenges to comprehend the connection between PD-L1 expression, EGFR status and ADAMTS8 - even in this figure the authors still ignore the clinical classification.
Author Response
Reviewer :
Comments and Suggestions for Authors
Lee et al are willing to investigate the role of ADMATS8 in the pathogenesis of lung cancer. The study has several experimental, intellectual and presentational weakness that contribute to the overall poor quality of the scientific product. Before commenting the content of the paper, we need to make some comments on the language, structure and figure quality of the manuscript because both interfere with the ability to comprehend what the authors are willing to say. To begin with, the use of English language throughout the manuscript is inadequate at best. There is a great need of language editing since it contains numerous grammatical errors and typos (way too many examples to add here) and in certain places is pretty unclear what the authors are willing to express.
Ans. Thanks for your detailed reviewing. Sorry for the spelling errors and typos. We have asked a native English speaker for grammar checks and typos. We have put more emphases on language editing.
Furthermore, the preparation and quality of the figures is low, blurry, overstretched and not clearly labeled throughout the manuscript that someone has to be constantly back and forth the figure legend in order to try to comprehend the message.
Ans. Thanks for your suggestions. We have improved the quality of these figures and labeled these figures clearly to avoid misunderstandings. We do believe these changes will not make the readers confused.
On a different note, the legends are not informative and repetitive. In addition, the structure of the paper with large gaps between the figures is very confusing and tiring to read. It seems like a collection of paragraphs and pages that non-coherently connect figures, making it difficult to follow.
Ans. Thanks for your suggestions. Based our figures, we have re-written the figure legends and results parts. We tried our best to have the coherent figure legends.
If someone can overcome these issues, that largely introduce increasing challenges on the understanding of the manuscript's content, there are several flows with the methodology itself. The authors constantly ignore, cherry-pick and move back and forth between different subtypes of lung cancer (LUAD, LUSC etc) without proposing or demonstrating why (figures 1,2,3 Given that these molecular/clinical subtypes are crucial for the prognosis and management of lung cancer patients, this is a severe flaw of the story, which is not justified within the text or the data.
Ans. Thanks for your detailed reviewing and suggestions. We have modified this article based on your suggestions. This article focused on the pathogenesis of lung adenocarcinoma. We have collected the several cohorts’ data of lung adenocarcinoma to possibly elucidate the importance of ADAMTS8. We have deleted the Oncomine cohorts irrelevant to lung adenocarcinoma in Figure 1E. Based on suggestion, we u
In figure 1F, the authors present data on the expression of ADAMTS8 in several lung cancer cells (ps in the methods for some reason they present PC-9, a lung adenocarcinoma cell line with EGFR mutation, as a lung cancer cell line, but the other cell lines as LUAD ones. The authors are not using widely used LUAD (A549, H1299, H1975 etc) or LUSC cell lines to support their claims. Furthermore, nowadays, there are several depositories with RNA-seq expression of several cancer cell lines (especially from the broad institute) that the authors could use, in order to provide more unbiased and robust data. In figure 3D,
Ans. Thanks for your comments. In this study, our focus is on the topic of lung adenocarcinoma. We utilized several cell lines as PC9 (adenocarcinoma, EGFR mutated), CL1-0 and CL1-5 (both are lung adenocarcinoma, Asian cell lines; they are isogenic and being to be an aggressive form as CL1-5). As we know that H1299 is not a typical adenocarcinoma cell line. We have removed the squamous cell carcinoma and large cell cohort in Figure 1E Based on suggestion, we took advantages of the Cancer Cell Line Encyclopedia (CCLE) database. There are two groups, one EGFR TKI sensitive (HCC827, HCC2279 and HCC4006) and one EGFR TKI resistant (NCIH1650 and NCIH1975). The results suggested that high expression of ADAMTS8 in EGFR TKI sensitive and low in EGFR TKI resistance (Figure 7D), which is consistent as Figure 7C. However, the relationship between EGFR TKI sensitive or not and ADAMTS8 is not clear but will be our further study aims.
I find extremely challenging to understand why the authors include data regarding GBM and other CNS tissues. Also in the version that I downloaded, figure 4 is missing.
Ans. Thanks for your concerns. The functions of ADAMTS8 was extracted from the public database as CancerSEA. Based on the availability of this dataset, it lacks any information about lung adenocarcinoma. This dataset provided the cellular functions of ADAMTS8 in GBM (glioblastoma multiforme), UM (uveal melanoma) and RB (retinoblastoma). Based on the results from CancerSEA, we provided the possible functions of ADAMTS8 in cancer. Further functional assays of ADAMTS8 in lung cancer will be our topic to pursue.
Moving on, data on Figure 5 demonstrate that GATA1 and LMO2 affect the prognosis of LUAD patients (again not defined why LUAD or other subtype), trying to make a connection with ADAMTS8 regulation. These data are purely non-informative and correlative. The fact that these widely regulatory TFs affect the prognosis has nothing to do with ADAMTS8 expression. To address this claim, the authors should perform a plethora of biological experiments with genetic knockdowns, mouse models etc.
Ans. Thanks for your concerns. The regulatory mechanism by transcription factors on ADAMTS8 was derived from the hTFtarget. Based on the criteria for collection, it offers the following functions for users to explore TF–target regulations. The GATA1 was acquired from it. The role of GATA1 for ADAMTS8 might be a co-association transcription factor.
We would like to perform a plethora of biologic functions and GATA1 regulatory mechanisms on ADAMTS8 in our future study.
In a similar trend, I have increasing challenges to comprehend the connection between PD-L1 expression, EGFR status and ADAMTS8 - even in this figure the authors still ignore the clinical classification.
Ans. Thanks for your suggestion. Our results are to imply the role of ADAMTS8 in non-actionable targets such as EGFR and PD1. Both actionable targets have the most powerful drugs as EGFR-TKIs and immune checkpoint inhibitors. Based on these, here we proposed that ADAMTS8 might play a role in lung cancer of non-actionable targets.
In addition, as mentioned before, we have deleted cohorts in Figure 1E which are poor response to EGFR TKI and ICIs, and we kept focuses on lung adenocarcinoma.
Reviewer 2 Report
Reviewer comments
Comment 1- Please provide the possible root causes of lung cancer.
Comment 2- Why lung cancer is among the leading causes of cancer death?
Comment 3- The authors have mentioned that new drug targets are needed for the treatment of lung cancer due to the low survival rate. What is the possible cause behind the low survival rate associated with targeted drugs?
Comment 4- The authors have explained the classification of ADAMTSs in a very excellent way. However, the authors did not mention the role of alternative splicing in the existence of multiple isoforms.
Comment 5- Do post-translational modifications of ADMTSs have any role in the biological functions and physiological actions of these metalloproteinases?
Comment 6- May the effects of single ADAMTS gene mutations in human Mendelian conditions be masked or modified by any strategy? Please take the help of previous publications.
Comment 7- It will be better to provide a graphical abstract for a better explanation of the NGS and Q-RT-PCR section.
Comment 8- Please mention the name of the tool for designing the primers used in this study.
Comment 9- Write brief importance of Next Generation Sequencing in the detection of diseases and their treatment strategies.
Comment 10- The expression of ADAMTS8 is shown to be reduced in lung cancer. The authors are suggested to explain the possible cause behind the reduction of its expression.
Comment 11- Please add figure 4 in the manuscript. Figure 4 is not shown. I think that it is a printing mistake.
Comment 12- This study shows that low expression of GATA 1 is linked with shorter survival time, and is correlated with low-level ADAMTS8. How does GATA1 regulate ADAMTS8? Any possible mechanism?
Comment 13- Please check the grammar. There are a few minor mistakes.
Author Response
Reviewer :
Comments and Suggestions for Authors
Reviewer comments
Comment 1- Please provide the possible root causes of lung cancer.
Ans. Thanks for reminding. We have added several root causes of lung cancer in our introduction as “Lung cancer has been a leading cause of cancer related death worldwide due to smoking, asbestos, radon exposure [1, 2]”.
Comment 2- Why lung cancer is among the leading causes of cancer death?
Ans. Thanks for your suggestion. Based on present scientific researches, the critical clinical issues for lung cancer not only in its pathogenesis but also its high recurrence rate. Despite the novel agents based on the tumorigenesis, recurrence of lung cancer is more common than other cancer type. The scientists and clinicians have devoted many efforts on lung cancer which improve the overall survival rate in the past decades. However, based on the data from the International Agency for Research on Cancer (IARC, https://gco.iarc.fr/today/online-analysis-multi-bars?v=2020&mode=cancer&mode_population=countries&population=900&populations=900&key=asr&sex=0&cancer=39&type=0&statistic=5&prevalence=0&population_group=0&ages_group%5B%5D=0&ages_group%5B%5D=17&nb_items=10&group_cancer=1&include_nmsc=0&include_nmsc_other=1&type_multiple=%257B%2522inc%2522%253Atrue%252C%2522mort%2522%253Atrue%252C%2522prev%2522%253Afalse%257D&orientation=horizontal&type_sort=0&type_nb_items=%257B%2522top%2522%253Atrue%252C%2522bottom%2522%253Afalse%257D), lung cancer is the third leading cause in 2020. That will be the reason why we are focusing on lung cancer research to aid patients.
Comment 3-The authors have mentioned that new drug targets are needed for the treatment of lung cancer due to the low survival rate. What is the possible cause behind the low survival rate associated with targeted drugs?
Ans. Thanks for your comments. The progression free survival for EGFR TKIs was only 9-17M. Based on the data about resistance (recurrence) of targeted therapy in lung cancer, the most common mechanism for druggable EGFR mutation is T790M mutation on exon 20 (Cai-HongYun, et al. 2008; PNAS; https://doi.org/10.1073/pnas.0709662105). The emergent other mechanisms such as c-Met amplification and HER2 amplification. This study is focusing on the possible new target for lung cancer treatment.
We have added it into the Text.
Comment 4- The authors have explained the classification of ADAMTSs in a very excellent way. However, the authors did not mention the role of alternative splicing in the existence of multiple isoforms.
Ans. Based on the data form NCBI_Gene, there are two splicing variants found. However, the cohort from UALCAN did not provide the RNA sequences from RNAseq. By this, we could not distinguish these two variants.
Comment 5-Do post-translational modifications of ADMTSs have any role in the biological functions and physiological actions of these metalloproteinases?
Ans. Thanks for your reviewing. The post-translational modification of ADAMTS8 (Salvatore Santamaria et al. 2021; JBC; doi: 10.1016/j.jbc.2021.101323) ADAMTS8 undergoes autolysis at six different sites within its spacer domain without knowing functions. We have added it in the Text as our reference. The posttranslational modifications might be worthy of further study.
Comment 6-May the effects of single ADAMTS gene mutations in human Mendelian conditions be masked or modified by any strategy? Please take the help of previous publications.
Ans. Thus, the effects of single ADAMTS gene mutations may be masked or modified in human Mendelian conditions by concurrent or compensating expression and functional overlap of a homolog, which then constitutes a key modifier gene. (Timothy J. Mead, 2018; Matrix Biol.; doi:10.1016/j.matbio.2018.06.002).
Comment 7-It will be better to provide a graphical abstract for a better explanation of the NGS and Q-RT-PCR section.
Ans. Thanks for your suggestion. We have added the graphic flow chart (Appendix Figure A1) for the NGS and qRT-PCR.
Comment 8-Please mention the name of the tool for designing the primers used in this study.
Ans. We have added it into the Materials and Methods section
Comment 9-Write brief importance of Next Generation Sequencing in the detection of diseases and their treatment strategies.
Ans. We have added the importance of NGS in detection diseases and treatment strategies as “Next-generation sequencing (NGS), a powerful tool, is developed in recent years to elucidate the mechanism of tumorigenesis and workout a possible solution with the aid of bioinformatics.”
Comment 10-Te expression of AD AMTS8 is shown to be reduced in lung cancer. The authors are suggested to explain the possible cause behind the reduction of its expression.
Ans. Thanks for your suggestion. To elucidate the possible regulatory mechanisms on ADAMTS8 would be critical to solve the issue of ADAMTS8 mediating lung cancer tumorigenesis. In this study, we tried several ways as DNA methylation, CNV, miRNA and transcriptional factors to possibly explain the regulatory mechanism. Here, we proposed that GATA1 might be play a role in regulating the expression of ADAMTS8 in Figure 5.
Comment 11- Please add figure 4 in the manuscript. Figure 4 is not shown. I think that it is a printing mistake.
Ans. Thanks for reminding. We have added Figure 4 into this article
Comment 12-This study shows that low expression of GATA 1 is linked with shorter survival time, and is correlated with low-level ADAMTS8. How does GATA1 regulate ADAMTS8? Any possible mechanism?
Ans. Thanks for your wonderful reviewing. Initially, we have the TFs for ADTAMTS8 from the hTFtarget (QiongZhang et al. 2020; Genomics, Proteomics & Bioinformatics; https://doi.org/10.1016/j.gpb.2019.09.006). The prediction GATA1 of ADAMTS8 was extracted. However, when having literature review, few or none study have shown the linked. Based on the principles for this site, the linkage between GATA1 and ADAMTS8 might be a co-association. The possible explanation might be that it was an indirect upstream regulation despite a good correlation in Table 1. More details and the exact mediators between GATA1 and ADAMTS8 and how it regulates ADAMTS8 would be worthy of analysis in our further study.
Comment 13-Please check the grammar. There are a few minor mistakes.
Ans. Thanks for your suggestions. We have a native English speaker for proof-reading.
Reviewer 3 Report
- In addition to qPCR, please use Western blot to verify the changes of ADAMTS8 protein expression level.
- Please use siRNA technology, knock down ADAMTS8 at cellular level and then compare the survival rate of cancer cells.
- The ordinate of Figure 1A and 1F is not very clear. It is suggested to replace to “Fold change of mRNA expression level”.
- Please add the meanings of *, **, and *** respectively in the figure legend of Figure 1.
- It is recommended that the language be modified and polished by professionals.
- Please correct errors in line 41 “posivetly” and in the ordinate of Supplement Figure 1 “velue”.
- In lines 171 and 223, please delete the extra Spaces.
- In line 150, please correct “form” to “from”.
- In lines 128 and 131, please correct unnecessary italics.
- Please add the catalog number and company name of each reagent used in the experiment, for example, in line 119 “SYBR Green”, in line 131 “RPMI 1640”, in line 135 “insulin”, etc.
- In line 186, “Statistical analysis” should be presented as a subheading 2.8 under Materials and Methods, rather than as a separate section.
- In line 390, “the results showed higher levels of ADAMTS8”, please check it again. Is it “higher”?
- In line 437, there is a missing period after the word “treatment”.
- The word size difference in a same figure should not be too big. For example, the word size in Figure 5A and B is a little small.
Author Response
Reviewer :
Comments and Suggestions for Authors
- In addition to qPCR, please use Western blot to verify the changes of ADAMTS8 protein expression level.
Ans. Thanks for your suggestion. The present study is to provide the possible association between ADAMTS8 and lung tumorigenesis. Based on present data from mRNA and public cohorts. We do have confidence on the role of ADAMTS8 in lung cancer. If possible, protein level would be more persuasive way and provide more confidence in it. However, due to time issue, we would not be able to have such data in this article. Our further investigation into lung cancer would performed such experiments.
- Please use siRNA technology, knock down ADAMTS8 at cellular level and then compare the survival rate of cancer cells.
Ans. Thanks for your suggestion. Based on the data we have, the expression of ADAMTS8 was low in cancer as in cell line as PC9, CL1-0, CL1-5 which is not suitable for knocking. The proper study design would be using overexpression, however, it takes months to complete it (at least 3M). We would like to do it in our further study.
- The ordinate of Figure 1A and 1F is not very clear. It is suggested to replace to “Fold change of mRNA expression level”.
Ans. Thanks for your suggestion. We have the quality improvement in these figure.
- Please add the meanings of *, **, and *** respectively in the figure legend of Figure 1.
Ans. Thanks for detailed reviewing. We have added *, **, and *** into our Figure 1.
- It is recommended that the language be modified and polished by professionals.
Ans. Thanks for your suggestion. We have an English speaker for typos and grammar errors check.
- Please correct errors in line 41 “posivetly” and in the ordinate of Supplement Figure 1 “velue”.
Ans. We have corrected it.
- In lines 171 and 223, please delete the extra Spaces.
Ans. We have corrected it.
- In line 150, please correct “form” to “from”.
Ans. We have corrected it.
- In lines 128 and 131, please correct unnecessary italics.
Ans. We have corrected it.
- Please add the catalog number and company name of each reagent used in the experiment, for example, in line 119 “SYBR Green”, in line 131 “RPMI 1640”, in line 135 “insulin”, etc.
Ans. We have added the necessary information for these products.
- In line 186, “Statistical analysis” should be presented as a subheading 2.8 under Materials and Methods, rather than as a separate section.
Ans. We have corrected it.
- In line 390, “the results showed higher levels of ADAMTS8”, please check it again. Is it “higher”?
Ans. We have rechecked and corrected it.
- In line 437, there is a missing period after the word “treatment”.
Ans. We have adjusted it.
- The word size difference in a same figure should not be too big. For example, the word size in Figure 5A and B is a little small.
Ans. Thanks for reviewing, we have corrected these errors.
Round 2
Reviewer 1 Report
Lee et al submit a revised version of their manuscript. This version is significantly improved compared with the originally submitted version. The authors thoroughly addressed all the comments largely improving the overall merit. Nevertheless, there are several scientific questions that the authors should address in their following studies.
The english has been definitely improved, allowing to understand the content of the manuscript adequately. The overall improvement in the methodology, cell lines and datasets used and the more defined scientific question of the manuscript, lead to an adequate scientific manuscript that worths of publication.
Congratulations to the authors for their efforts.
Reviewer 3 Report
In the revised version, the authors have answered my questions and made appropriate modification to the manuscript.